# Mesenchymal Stem Cell-Derived Extracellular Vesicles for Osteoarthritis Treatment: Extracellular Matrix Protection, Chondrocyte and Osteocyte Physiology, Pain and Inflammation Management

**DOI:** 10.3390/cells10112887

**Published:** 2021-10-26

**Authors:** Thu Huyen Nguyen, Chau Minh Duong, Xuan-Hung Nguyen, Uyen Thi Trang Than

**Affiliations:** 1Department of Bioscience, University of Milan, 20133 Milan, Italy; thuhuyen.nguyen@studenti.unimi.it; 2Vinmec Research Institute of Stem Cell and Gene Technology, Vinmec Healthcare System, Hanoi 100000, Vietnam; cduong@clarku.edu (C.M.D.); v.hungnx1@vinmec.com (X.-H.N.); 3Department of Biology, Clark University, Worcester, MA 01610, USA; 4Vinmec Research Institute of Applied Sciences and Regenerative Medicine, Vinmec Healthcare System, Hanoi 100000, Vietnam; 5College of Health Sciences, VinUniversity, Hanoi 100000, Vietnam

**Keywords:** osteoarthritis, extracellular vesicles, mesenchymal stem cells, chondrocytes, inflammation, bone homeostasis

## Abstract

Osteoarthritis (OA) is a common degenerative disease that can lead to persistent pain and motion restriction. In the last decade, stem cells, particularly mesenchymal stem cells (MSCs), have been explored as a potential alternative OA therapy due to their regenerative capacity. Furthermore, it has been shown that trophic factors enveloped in extracellular vesicles (EVs), including exosomes, are a crucial aspect of MSC-based treatment for OA. Evidently, EVs derived from different MSC sources might rescue the OA phenotype by targeting many biological processes associated with cartilage extracellular matrix (ECM) degradation and exerting protective effects on different joint cell types. Despite this advancement, different studies employing EV treatment for OA have revealed reverse outcomes depending on the EV cargo, cell source, and pathological condition. Hence, in this review, we aim to summarize and discuss the possible effects of MSC-derived EVs based on recent findings at different stages of OA development, including effects on cartilage ECM, chondrocyte biology, osteocytes and bone homeostasis, inflammation, and pain management. Additionally, we discuss further strategies and technical advances for manipulating EVs to specifically target OA to bring the therapy closer to clinical use.

## 1. Introduction

Osteoarthritis (OA) is an age-related degenerative joint disorder resulting from pre-existing joint abnormalities or risk factors, including age, female sex, obesity, anatomical characteristics, muscle weaknesses, and joint injury [1]. According to the Global Burden of Disease Study (http://ghdx.healthdata.org/gbd-2019, accessed on 25 April 2021), OA is considered the most common form of arthritis worldwide, and this rapidly increasing health condition affects 7% of the population globally, leading to debilitating symptoms [2]. As a result of an aging population and expanding obesity epidemic phenomenon, by 2019, OA became the fifteenth-highest leading cause of disability [2]. Although OA can occur in any synovial joint in the body, it most commonly affects the hands, knees, and hips, resulting in a severe reduction in quality of life as well as burdens on the patient’s family and society [3].

During OA progression, pro-inflammatory and inflammatory mediators released by immune cells and extracellular matrix (ECM) degradation trigger immunomodulatory processes. The immune system is stimulated to produce pro-inflammatory cytokines, including tumor necrosis factor-alpha (TNF-α), interleukin (IL)-1, IL-6, IL-2, IL-15, and IL-10 [4], along with proteases, such as matrix metalloproteinase (MMP)-1, MMP-3, MMP-13 (collagenase), and a disintegrin and metalloproteinase with thrombospondin motifs 5 (ADAMTS-5) (aggrecanase) [5]. Consequently, these factors result in structural degradation of cartilage ECM components, damage to subchondral bone and ligaments, and joint capsule hypertrophy [6]. Abnormal remodeling of the subchondral bone architecture via osteoclast-mediated bone resorption and osteoblast-mediated bone formation is a hallmark of OA and results in hypomineralization, loss of stiffness, and bone thickening [7].

The treatment of OA has always been challenging. Currently, there is still no gold-standard treatment available for OA, and therapeutic approaches aim primarily to manage disease-associated pain rather than address the underlying cause. Current OA therapies are categorized into nonpharmacological, pharmacological, and surgical approaches [8]. Nonpharmacological therapies, such as exercise, weight loss, and physical therapy, only improve the functional conditions of early-stage OA patients [2]. Although pharmacological agents, particularly acetaminophen, nonsteroidal anti-inflammatory drugs (NSAIDs), opioids, topical analgesics, corticosteroids, and hyaluronic acid, reduce pain and improve joint mobility [8], they have negligible or no influence on the repair of damaged cartilage and restoration of cartilage homeostasis. Surgical arthroplasty is still most widely used for end-stage OA patients with severe functional disabilities despite the risks for infection, thrombus formation, and secondary surgery, especially in elderly patients [8]. In recent years, cell-based therapies, including autologous chondrocyte- or stem cell-based approaches, have emerged as a promising option for OA treatment [9]. Although autologous chondrocytes appear to be a safe and productive solution, their limited availability and dedifferentiation capacity and loss of function during in vitro expansion limit their application as an OA therapy [9]. Other stem cells, particularly mesenchymal stem cells (MSCs), have become an attractive cell type for repairing damaged OA joints; however, many questions remain unanswered regarding the safety of MSCs, their homing capacity, and their mechanisms of action [10]. Therefore, MSC-derived extracellular vesicles (EVs), in particular exosomes, are emerging as a novel and effective cell-free alternative to MSC therapy against numerous disease targets, including OA [11].

## 2. MSC-Derived EVs as Potential Cell-Free Therapy for OA

MSCs have been widely used as alternative cell sources for OA treatment through direct intra-articular injection [12]. However, there are concerns associated with MSC preparation, such as genetic instability and chromosomal alteration during long-term culture ex vivo and the increased immunogenicity of differentiated cells; therapeutic efficacy, including low engraftment and the proportion of MSCs that can reach target tissues is another concern, as is alloantibody production caused by repeated administration [10]. Thus, EVs, including microvesicles and exosomes, have been proposed to replace conventional cell-based therapy for OA treatment due to advantages including (1) better safety profiles and fewer side effects due to the natural lipid and surface protein composition [13]; (2) lower immunogenicity [13]; (3) protection of therapeutic agents (nucleic acids and proteins) from degradation and targeting to treatment sites [14,15]; (4) ability to cross some biological barriers that MSCs cannot [14]; and (5) easier preservation methods and fewer ethical issues [15]. In OA-related studies, exosomes have been isolated from MSCs derived from different sources, including bone marrow, adipose tissue, umbilical cord, synovial membrane/fluid, embryonic stem cells, or induced pluripotent stem cells [13]. Tissue origin has been demonstrated to affect EV potential; for example, adipose-derived MSC (ADMSC)-derived EVs showed a higher ability to induce cartilage and bone regeneration than bone marrow-derived MSC (BMMSC)- and synovial MSC (SMSC)-derived EVs [16].

The composition of MSC-derived exosomal cargoes, including RNAs, proteins, lipids, and metabolites, which determine the effectiveness of exosomes, may vary depending on cellular origins, developmental phases, epigenetic modifications, and stimulation conditions [13]. Recently, the potential role of glycans, which consist of a large number of monosaccharides linked through glycosidic bonds, in EVs has attracted interest from investigators. This type of molecule, which has been well reviewed by Della Rosa et al. and William et al. [17,18], contributes to various biological processes, ranging from biogenesis to functions such as therapeutic activity and cell targeting. Due to the ability to attach to the protein on the surface of EVs, this glycan is promising for engineering EVs toward precise targets and developing a delivery system that sustains EVs to improve their performance. However, while MSC-derived exosomal glycans can enhance osteogenic differentiation, knowledge of their application and involvement in OA is lacking [19]. Thus, more studies in the field of MSC-derived exosomal glycans, especially in relation to OA biology, are required.

MSC-derived exosomes promote the repair and regeneration of cartilage in the OA model by different mechanisms, such as: (1) enhancing matrix synthesis and preventing cartilage destruction; (2) promoting chondrocyte proliferation and migration while suppressing apoptosis; and (3) affecting immunomodulatory signals [20,21,22]. For example, ADMSC-derived EVs (exosomes and microvesicles) can rectify abnormal osteoblast metabolism and promote cartilage and bone regeneration, both in vitro and in vivo, by regulating focal adhesion, ECM-receptor interaction, actin cytoskeleton, cAMP, and PI3K-Akt signaling pathways [16,23]. Moreover, BMMSC-EVs were internalized rapidly by OA chondrocytes, abrogated TNF-a adverse effects in OA chondrocytes, and efficiently promoted cartilage regeneration in vitro [24]. The effects of MSC-derived EVs on different OA biological processes could be obtained by delivering a large number of proteins (Table 1) or miRNAs [25]; for example, exosomal miRNAs from amniotic membrane MSCs also exert chondroprotective and renoprotective effects by suppressing inflammation-related genes [26]. miR-26a-5p-enriched exosomes from BMMSCs repress the damage to synovial fibroblasts via PTGS2 in an OA rat model [27]. Notably, MSC-derived exosomes have been postulated to promote the differentiation of hMSCs toward chondrogenesis [20,28]. This is an important clue for the development and application of exosomes as an alternative cell-free therapy for OA treatment. Despite the substantial potential of MSC-derived EVs, there are many obstacles to the clinical use of this approach that will be further discussed in the next sections.

## 3. Factors Stimulating MSCs to Secrete Therapeutic EVs for OA

Depending on their biogenesis mechanism, EVs can be classified into three categories: apoptotic bodies, microvesicles, and exosomes. Among these, exosomes are the most attractive to researchers, followed by microvesicles, and there is not much consideration of apoptotic bodies, which are apoptosis products. The formation of exosomes starts from the inward budding of an endocytic vesicle that is then fused with early endosomes in the cytoplasm. This early endosome then develops into multivesicular bodies. Multivesicular bodies either deliver their contents to the lysosome for degradation or release exosomes into extracellular spaces by fusing their membrane with the cell membrane [34]. Although the factors involved in determining the fate of multivesicular bodies have not been well investigated, several studies have shown that cholesterol-rich MVBs fuse to the plasma membrane. Additionally, the endosomal sorting complex required for transport (ESCRT)-dependent pathway or ESCRT-independent pathway has been proposed to contribute to regulating exosome biogenesis [35]. In a different mechanism, microvesicles are formed by outward budding and fission of the plasma membrane. The direct budding of microvesicles occurs only at plasma membrane sites with changes in local lipid composition and phospholipid translocation. This process is completed through ARF6 and RHOA-dependent actin cytoskeleton reorganization [34].

Notably, exosome formation/release and the accumulation of molecules in exosomes depend on various factors. These may be different chemical, environmental and mechanical stimulants, such as gamma irradiation, hypoxia, acidosis, and matrix detachment. The tissue origin, such as bones, muscles, or brain, and healthy or pathological states of the secreting cells also influence molecules packaged into exosomes. For example, senescent cells can secrete exosomes carrying different contents, such as a low level of miR-140-3p, leading to exosomes with impaired regenerative capacity compared to those secreted by nonsenescent cells [36,37].

To date, researchers have maintained MSCs in different culture conditions to obtain the therapeutic effects of EVs on OA physiology, including conventional culture media, pro-inflammatory stimuli, specific culture systems, and even modified MSCs. Traditional media for MSC expansion, including DMEM or α-MEM supplemented with FBS or platelet lysate, have been widely used. For example, BMMSCs cultured in α-MEM supplemented with 5% human platelet lysate and heparin could release exosomes that exposed their capacity to inhibit inflammatory mediators of cartilage homeostasis and promote proteoglycan and COLII production, leading to cartilage regeneration in vitro [24]. Additionally, He et al. and Li et al. cultured BMMSCs in DMEM supplemented with FBS to release therapeutic exosomes to protect against cartilage degradation and relieve knee pain in rat OA models [38,39]. A pain relief effect may be achieved due to the abrogation of aberrant GGRP-positive nerve invasion in the subchondral bone [38]. The cartilage protection effect may be due to exosomes upregulating COL2A1, iNOS, and CGPR and downregulating MMP-13, consequently promoting cartilage (hyaline cartilage) repair in OA rats [39]. Culturing different cells under similar culture conditions can produce different effects; for example, ADMSCs cultured in DMEM or α-MEM supplemented with 10% FBS released exosomes enriched with TIMP-1, TIMP-2, and microRNAs, resulting in PGE2 expression, MMP activity, and regulation of the main pathways associated with synovial inflammation [40,41].

Chemical stimulation using pro-inflammatory cytokines such as IFNγ, TNF-α, and IL-1β has also been used to generate human MSC-derived exosomes with therapeutic effects on OA [42,43,44]. Stimulating ADMSCs with IFNγ and TNF-α produced exosomes carrying miR-34a-5p and 146a-5p that were then delivered to macrophages and induced a shift from the inflammatory M1 phenotype to the anti-inflammatory M2 phenotype [42], while stimulating ADMSCs with IFNγ, TNF-α, and IL-1β produced exosomes carrying microRNAs (miR-24-3p, miR-222-3p and miR-193b-3p), matrix metalloproteinase inhibitors (TIMP1, TIMP2), plasminogen, and cathepsin S to drive anti-inflammatory M2 polarization from the inflammatory M1 phenotype and reduce matrix degradation activities [44]. Moreover, MSCs could be transfected with lentiviral vectors overexpressing KLF3-AS1 or miR-140-5p to generate exosomes with the capacity to increase COL2A1 and aggrecan and decrease MMP-13 and RUNX2 in chondrocytes, induce chondrocyte proliferation and migration, and inhibit apoptosis [21,45]. In addition, researchers have used different culture systems; for example, UCMSCs maintained in rotary cell culture systems produce lncRNA H19-bearing exosomes that increase chondrocyte proliferation and matrix synthesis, inhibit apoptosis, and relieve pain [46]. The aforementioned examples indicate the inconsistent culture conditions and variety of methods used to generate MSC-derived EVs among research groups worldwide. This is due to the development of techniques for obtaining EV vesicles for a wide range of applications and the use of different stimulating factors to obtain specific therapeutic EV products.

## 4. Animal Osteoarthritis Models for EV Investigation

Several animal models have been developed for the evaluation of MSC-derived exosomes for OA in vivo. These models involve chemically/surgically induced or naturally occurring/genetically modified spontaneous OA [47]. Surgical models involve using a drill or punch tool to create the defect, mainly in the femoral trochlear grooves [48,49,50] or knee [51,52], or to cut the collateral ligament and meniscus without damaging the tibial surface [45]. Li et al. established lumbar facet joint osteoarthritis in a mouse model by surgery [38]. The results are highly reproducible and the disease progresses rapidly, making surgical models an excellent choice for short-term studies, including following the early stages of OA development and measuring the effect of early drug treatment [53]. While creating the defect by surgery is more common, intra-articular injection of chondrotoxic or pro-inflammatory substances such as collagenase and MIA (monosodium iodoacetate) into the knee joint also mimics the OA phenotype. Collagenase induces the degeneration of articular cartilage by directly digesting collagen from the extracellular matrix of cartilage [54] and articular instability by increasing joint laxity [55,56]. MIA is a metabolic inhibitor that breaks down the cellular aerobic glycolysis pathway and consequently induces cell death by inhibiting the activity of glyceraldehyde-3-phosphate dehydrogenase in chondrocytes [53]. The intra-articular injection of MIA leads to a reduction in the number of chondrocytes and, consequently, histological and morphological articular alterations, similar to the changes observed in human OA [57,58]. Despite the appropriateness of using all these factors, collagenase (type II or VII) is more commonly used to induce the OA mouse model [20,21,59,60], while MIA is preferred to induce the OA rat model [22,31,61].

Although in vivo evaluation has been conducted widely in small animal models, including rabbits, rats, and mice, clinical translation requires the validation of MSC-derived exosome safety and efficacy to repair and regenerate cartilage lesions in large appropriate animal models, such as horses, pigs, or dogs. Indeed, compared to small animals, large animals have more similar bone development to that in humans, i.e., they have closed growth plates at skeletal maturity, and OA in large animal models occurs more slowly than in small animal models, mimicking natural human disease [62]. However, based on our up-to-date findings, no preclinical studies have been carried out on large animals to test MSC-derived EVs, probably due to a lack of models with spontaneous early-onset OA; instead, large animals develop OA naturally with age.

## 5. MSC-Derived EV Promotes the ECM Regeneration

Changes in ECM composition and structure are characteristics of OA. Collagen type II and aggrecan—two major components of the extracellular matrix of articular cartilage—contribute to a healthy cartilage matrix; thus, degrading cartilage ECM proteins leads to the loss of cartilage integrity [5]. During OA progression, the increased expression of collagenases, such as matrix metalloproteinase 13 (MMP-13) and two aggrecanases, including disintegrin and metalloproteinase with thrombospondin motifs (ADAMTS)-4 and -5, results in a decrease in collagen type II and aggrecan [63]. MSC-derived exosomes can reverse ECM degradation by increasing the expression level of matrix proteins and other chondrogenic-related genes while reducing matrix degradative enzymes (Table 2). Exosomes can also enhance glycosaminoglycan synthesis and uniform glycosaminoglycan distribution [61]. Interestingly, exosomes originating from human umbilical cord Wharton’s jelly MSCs improved the reparative efficacy of the acellular cartilage extracellular matrix scaffold in an osteochondral defect rabbit model by promoting the secretion of cartilage matrix [64]. Weekly intra-articular injection of 100 μL human embryonic MSC-derived exosomes (concentration of 1 μg/μL) each up to 12 weeks was also able to enhance matrix synthesis of collagen type II and sulfated glycosaminoglycans (s-GAG) [65]. After 12 weeks of exosome treatment, defects showed good regeneration, with hyaline cartilage integrated completely with the surrounding cartilage and subchondral bone closely matched to the age-similar native control [65]. Moreover, EV treatment can also induce chondrogenic differentiation in vitro by BMMSCs with similar ECM changes in chondrocytes, such as increased expression of aggrecan, collagen type II, Sox9, and GAG. Among three MSC sources, bone marrow, adipose tissue, and synovium, ADMSC-derived EVs showed the most potential for promoting chondrogenic differentiation based on changes in the levels of ECM components [16].

Exosomes may mediate OA ECM through several mechanisms; for example, exosomal CD73 triggers adenosine activation of AKT/ERK and the AMPK pathway. The AKT/ERK pathway plays an important role in cellular survival, proliferation, and matrix synthesis, including those of chondrocytes, and the AMPK pathway is involved in matrix homeostasis [28,48]. UCMSC-derived exosomes induce TGF-β expression, which regulates the synthesis of ECM components, including aggrecan, proteoglycan, and type II collagen, by chondrocytes [22,61] and activates the TGF-β-dependent Smad2/3 signaling pathway, which facilitates cartilage repair in vivo [66]. ADMSC-derived EVs were abundant with versican, which is a chondroitin sulfate proteoglycan that promotes chondrogenesis and joint morphogenesis [41,67]. Additionally, EVs from human adipose-derived stem cells highly express tissue inhibitors of metalloproteinase-1 and -2 (TIMP-1 and TIMP-2) that control MMP activities and may play an essential role in matrix regulation [22]. Proteomics analysis of EVs derived from ADMSCs, BMMSCs, and SMSCs revealed that proteins packed in EVs upregulate the ECM organization process by inducing focal adhesion and ECM-receptor interaction, and ADMSC-derived EVs have the highest abundance of related proteins among the three tested sources [16].

Recently, many researchers have evaluated the underlying mechanisms of MSC-derived exosomal miRNAs in ECM modulation (Table 3) [22,28,51,52]. The effectiveness of ECM protection is due not only to exosomal microRNAs but also to exosomal proteins. For instance, synovial mesenchymal stem cell (SMSC)-derived exosomes carry miR-140-5p, which inhibits RalA to rescue SOX9 expression, leading to a restoration of ECM secretion. These SMSC-derived exosomes also carry Wnt5a and Wnt5b to activate YAP via the alternative Wnt signaling pathway, consequently promoting articular chondrocyte proliferation and migration [45]. The mechanism by which these exosomal miRNAs mediate cartilage ECM has yet to be fully understood, and further investigation is required to develop therapeutic exosome products against joint cartilage degradation. ECM damage is an important event that begins in early-stage OA and contributes to more severe OA development if not addressed; therefore, more investigations are required to validate the aforementioned results and to identify the pathways involved in catabolic and anabolic processing of matrix cartilage.

## 6. MSC-Derived EVs Inhibit Apoptosis and Promote the Migration and Proliferation of Chondrocytes

In the context of OA, the induction of chondrocyte apoptosis is associated with cartilage degradation and OA progression [73] and can be prevented by exosomes [48,66]. Interestingly, exosomes can inhibit apoptosis by inducing anti-apoptosis-associated gene expression. For instance, exosomes from human embryonic stem cell-derived MSCs enhance the expression of anti-apoptosis genes such as *Survivin* and *Bcl-2* while reducing CCP3+ apoptotic cells in an osteochondral-defective rat model [48]. In a specific temporomandibular joint OA model, hESC-derived exosomes significantly decreased the number of CCP3+ apoptotic cells and downregulated the expression of *Bax* genes [28,74]. The *Bax/Bcl-2* ratio, an important apoptosis marker, was reduced [74], and protein and mRNA levels were significantly decreased in OA chondrocytes following treatment with lncRNA H19-carrying exosomes [27,46,75]. In addition, exosomes derived from BMMSCs and hMSCs can induce the downregulation of p38, MAPK, ERK and activate the AKT pathway [48,76]. These exosomes can also rescue the p38-mediated mitochondrial apoptotic pathway by restoring the mitochondrial membrane potential [73,76].

Exosomal RNAs may also induce anti-apoptotic effects. For example, UCMSC-derived exosomes containing lncRNA H19 act as a competitive endogenous RNA to block miR-29b-3p, enhancing FoxO3 expression. Subsequently, FoxO3 promotes cartilage development, inhibiting chondrocyte apoptosis in vitro [50]. Additionally, MSC-derived exosomes carrying lncRNA KLF3-AS1 can interrupt the interaction between miR-206 and G-protein-coupled receptor kinase interacting protein-1 (GIT1), leading to an increase in GIT1 expression [21]. The abundant amount of exosomal miR-100-5p derived from infrapatellar fat pad MSCs can inhibit mTOR protein production, resulting in autophagy [52]. Both of these processes, in turn, impede chondrocyte apoptosis. Overall, exosomes have potential therapeutic effects against OA as they can deliver therapeutic molecules associated with different signaling pathways that indirectly inhibit chondrocyte apoptosis.

Chondrocyte migration and proliferation are two essential processes for maintaining healthy cartilage, and both are downregulated in OA. Interestingly, exosomes from various sources were found to promote the proliferation, migration, and viability of osteoarthritic chondrocytes in a dose-dependent manner [21,22,24,48,76]. For instance, the higher the dose of exosomes, the earlier proliferation was observed, and only the 10 μg exosome dose was sufficient to promote chondrocyte migration [48]. Additionally, several researchers have investigated proteins associated with chondrocyte adhesion, migration, and proliferation under the regulation of MSC-derived exosomes [44,68]. Exosomes trigger changes in gene expression, such as FGF-2, survivin, and Bcl2/Bax, to regulate cellular proliferation [48,66] or reverse the inhibitory effects of TNFα and IL1β on cell migration and cell proliferation [24,69]. Furthermore, enhanced proliferation was explained by the activation of the TGFβ1 and Smad2/3 signaling pathways triggered by exosomes derived from UCMSCs [66] or the activation of YAP with Wnt5a and Wnt5b by exosomes extracted from synovial MSCs [45]. In the TMJ-OA model, hMSC-derived exosomes alleviated the decrease in proliferation and migration by increasing s-Gag synthesis and suppressing NO and MMP13 production in matrix homeostasis. Exosomal CD73 was proposed to play an important role in the underlying molecular mechanism by activating AKT, ERK, and AMPK phosphorylation [31]. With increasing GAG production and COL II protein expression, cartilage regeneration in chondrocytes was also enhanced by EVs from BMMSCs, ADMSCs, and SMSCs, among which ADMSC-EVs produced the most significant outcome [16]. Moreover, EV proteins from these sources significantly regulate ECM homeostasis and actin cytoskeleton dynamics, which indirectly contribute to chondrocyte proliferation and migration [16].

Additionally, MSC-derived exosomes are able to regulate cell proliferation and migration by transferring genetic material, such as miRNA [20,44,45,52]. This regulation can be achieved through different mechanisms; for example, exosomal miR-23a-3p could activate the PTEN/AKT signaling pathway and promote cell migration in vitro [51], and miR-26a-5p-overexpressing exosomes from BMMSCs could promote the proliferation of synovial fibroblasts by downregulating PTGS2, which was confirmed in an OA rat model [27]. Moreover, UCMSC-derived exosomal lncH19 significantly increased the number of transmembrane chondrocytes by sponging miR-29b-3p to upregulate Fox3 and inhibiting OA-induced signaling molecules, leading to a promotion of chondrocyte proliferation and migration [46,50,77]. Furthermore, exosomal lncH19 derived from other sources, such as fibroblast-like synoviocytes, was shown to interfere with miR-106b-5p and promote the production of TIMP2, thus maintaining cell proliferation ability [77]. Therefore, genetic materials, especially miRNAs, carried by exosomes could be used as potential therapeutic agents for OA treatment to regulate chondrocyte biology (Figure 1).

A question has been raised regarding how EVs can penetrate cartilage tissue to perform reparative functions. Chondrocytes are encased in a dense, negatively charged, and avascular extracellular matrix and EV permeation through the matrix to interact with chondrocytes might be challenging. Therefore, studies of MSC-derived EVs for use in early-stage OA may focus on superficial chondrocytes and cartilage matrix maintenance. To reach chondrocytes at the deeper layer, a chondrocyte-affinity peptide should be engineered on the exosome surface; this facilitates exosome penetration into the middle zone of the cartilage tissue to target chondrocytes for better therapeutic efficacy [78]. Alternatively, during the latter period of the disease, inflamed chondrocytes may exhibit chemoattractant receptors to particular EVs, for example, to neutrophil-derived EVs but not those released from monocyte-derived macrophages [79]. These neutrophil-derived EVs enter the cartilage and bind the FPR2/ALX receptor, inducing TGF-β1 production and ECM deposition while protecting chondrocytes from apoptosis [79]. However, the specific mechanism by which neutrophil-derived EVs enter cartilage is poorly understood.

## 7. MSC-Derived Exosomes Reduce Inflammation

Exosomes can reduce inflammation and protect healthy cartilage from inflammatory inducers in various ways [26,28,80]. This effect may be due to exosomes increasing M2 macrophage polarization and decreasing M1 macrophage production in osteoarthritic models [26,80]. The imbalance of M1/M2 macrophages is an important factor when assessing OA progression since M1 macrophages produce pro-inflammatory effects, while M2 macrophages produce anti-inflammatory effects. In an OA model, the activation of M1 macrophages worsens inflammation and cartilage damage, while the repair and remodeling activity induced by M2 macrophages is reduced [81]. Interestingly, dual regulation of macrophage phenotypes arises from exosomal microRNAs; for example, miR-27a-3p, miR-27b-3p and miR-130a-3p promote M1 macrophages and suppress M2 macrophages, while miR-24-3p, miR-146a-5p, miR-222-3p, miR-34a-5p, and miR-181a-5p promote M2 macrophage production and polarization and miR-24-3p miR-146a-5p enhance the M1 macrophage phenotype [26]. Macrophage polarization occurs due to these exosomal microRNAs targeting TLR4 through NF-κb, subsequently suppressing STAT3 to increase M2 polarization [44]. Notably, synovial M1-polarized macrophages, which are reduced by exosomal miRNAs, secrete pro-inflammatory factors, while M2-polarized macrophages, which are promoted by exosomal miRNAs, ensure cartilage graft survival and support healthy cartilage; this results in a reduction in inflammation and an increase in joint repair in vivo [82]. Thus, exosomal miRNAs could be potential candidates for osteoarthritis treatment by reducing inflammation and promoting cartilage regeneration.

Moreover, a reduction in inflammation via MSC-derived exosomes could be obtained by inhibiting the expression of pro-inflammatory factors while enhancing the expression of anti-inflammatory factors (Figure 2). For example, exosomes reduced the expression of pro-inflammatory interleukins, such as IL-1A, IL-1B, IL-6, IL-8, and IL-17, in OA chondrocytes [24,27,69] but also triggered the production of anti-inflammatory IL-10 [23]. The effects of EV/exosomes on the expression of anti-inflammatory and inflammatory factors were not limited to reducing chondrocyte apoptosis and enhancing chondrogenesis, as osteoblast senescence was also reduced [23], promoting osteochondral and hyaline cartilage regeneration in an animal model [64]. Thus, this aspect of OA inflammation modulation by exosomes is important and requires more investigation into the underlying mechanism to promote the development of therapeutic exosomes for specific disease processes.

## 8. MSC-Derived Exosomes Regenerate Osteocyte Physiology and Bone Regeneration

Osteoarthritis progression causes adverse effects on osteocyte physiology, including cartilage degradation and subchondral bone sclerosis [83]. Significantly, the two critical processes of bone formation and bone resorption are interrupted during OA due to the misregulation of osteoblast and osteoclast activity responsible for adjusting bone matrix synthesis and degradation [83]. This phenomenon causes abnormal bone remodeling, which in turn triggers further OA damage [83]. Meanwhile, EVs show great therapeutic potential for regenerating normal osteocyte physiology and alleviating subchondral bone deterioration in different OA models [48,61,68]. In an immunocompetent rat model of temporomandibular joint osteoarthritis, MSC exosomes attenuated bone loss after two weeks and restored subchondral bone integrity after eight weeks [28,60]. Additionally, in the lumbar facet joint (LFJ) of the OA model, treatment with 200 µg BMMSC exosomes consecutively four weeks after surgery blocked subchondral bone erosion and decreased cartilage degeneration by suppressing the RANKL/RANK/TRAF6 pathway [38]. Moreover, the protective effect on bone is mediated not just by exosomes but also by microvesicles. Cosenza et al. showed that increasing the concentration of microvesicles generated from murine BMMSCs to twice that used with exosomes (500 ng/5 µL vs. 250 ng/5 µL) produced a capacity equivalent to that of exosomes to reduce bone degradation in epiphyseal and subchondral bones in an OA mouse model [60].

In addition to the bone remodeling process, exosomes promoted cartilage restoration in the LFJ-OA model and mono-iodoacetate (MIA)-induced OA model [38,61]. After six weeks of exosome treatment with 100 µg AF (amniotic fluids) MSC-derived exosomes, the MIA-induced OA model showed an increase in neotissue filling covering the damage on the joint surface and good surface regularity [61]. Thus, exosomes can affect osteocyte physiology, induce bone remodeling and prevent cartilage degeneration, subsequently halting OA damage.

## 9. MSC-Derived Exosomes May Relieve OA Pain by Modulating Inflammation and Cartilage Matrix Function

Joint pain in OA conditions results from the deterioration and damage of cartilage, tendon and ligament stretch, and even bone rubbing at the severe disease stage [1]. Uncontrolled inflammation can be an active pathway involved in the pathogenesis of OA pain. Additionally, chronic pain can lead to loss of mobility and psychological disorders such as anxiety, depression, and insomnia [1].

As discussed, exosomes have been investigated for their roles in many physiological aspects of OA, but investigations into their use for pain management are lacking. There have been several preliminary studies in which EV treatment reduces pain in OA models [38,39]. He et al. injected 40 μg BMMSC-derived exosomes intra-articularly into a sodium iodoacetate-induced OA mouse model and then assessed pain scores by evaluating mechanical paw withdrawal threshold (PWT) and thermal paw withdrawal latency (PWL). Higher PWT and PWL values were obtained in OA mice treated with exosomes compared with untreated OA mice, indicating that BMMSC-derived exosomes can relieve OA pain [39]. In another lumbar facet joint (LFJ) osteoarthritis model created by Li et al., 200 μg of BMMSC-derived exosomes was injected into the tail vein weekly for four weeks. Then, pain behaviors were determined using the vocalization threshold test in LFJ osteoarthritis animals. Analgesic effects of BMSMC-derived exosomes were observed at week four of the treatment as evidenced by an increase in the vocalization pressure threshold [38]. In addition to reducing pain in OA, exosomes from plasma and macrophages have also been reported to have the capacity to relieve pain in other chronic pain conditions, such as complex regional pain syndrome (CRPS) [84] and an inflammatory pain model caused by complete Freund’s adjuvant [85]. Analgesic effects of exosomes may be obtained from exosomal content delivery to change the expression of inflammatory factors at the pain site. For example, exosomal miR-338-5p could bind to IL-6 mRNA and then inhibit IL-6 translation by targeted cells; consequently, the low level of IL-6 in plasma contributes directly to a reduction in inflammation, leading to pain alleviation [84]. Additionally, exosomes reduced inflammation and thermal hypersensitivity in an inflammatory model and restored normal sensitivity via an unknown molecular pathway [85]. Exosomes have also been found to decrease anabolic factors, such as COL2A1 and ACAN, and increase catabolic factors, including MMP13 and ADAMTS5, to promote cartilage matrix repair in the OA model, resulting in pain alleviation [39]. Therefore, controlling inflammation and cartilage function may be beneficial mechanisms to relieve OA patient pain. These are preliminary data on the use of MSC-derived exosomes for OA pain relief, and the mechanism remains poorly understood.

## 10. Strategy to Develop Therapeutic EVs

As mentioned, stem cell-derived exosomes carry many bioactive molecules, such as proteins, lipids, and genetic materials, that can be used in therapeutic, prognostic, and diagnostic methods. Interestingly, specific molecules can be loaded into exosomes by either endogenous or exogenous loading techniques (Figure 3) [86]. For example, Gong et al. loaded miR-30b specifically into exosomes using transfection of MSCs with lentiviral particles carrying the premiR-30b fragment, leading to a 5.22-fold increase in accumulation of miR-30b in exosomes derived from transfected cells compared to those derived from control cells [87]. The miR-30b-enriched exosomes were then applied to human umbilical vein endothelial cells and increased tube length by nearly 68% over that observed in the control [87]. Additionally, Tang et al. designed a protocol to transfer a plasmid coding for IL-10 into RAW cells, resulting in a 7-fold increase in the secretion of IL-10-enriched exosomes compared to those secreted by the control [88]. IL-10+ exosomes target tubular endothelial cells (TECs), reduce cell apoptosis, alleviate tubular injury and promote renal recovery by maintaining mitochondrial fitness in TECs [88]. Given the strategies above, endogenous loading of trophic molecules into exosomes can be achieved by modifying parental cells using genetic engineering or incubation with cytotoxic drugs [89]. In terms of exogenous loading, molecules are encapsulated into naturally secreted exosomes in vitro through incubation, chemical transfection, electroporation, sonication, extrusion, or freeze-thaw cycles, as reviewed by Tang et al. [89]. 

For instance, an LNA (locked nucleic acid)-modified anti-miR-142-3p oligonucleotide was transfected successfully into BMMSC-derived exosomes using the electroporation method [90]. Exosomes carrying LNA-anti-miR-142-3p suppressed the expression levels of miR-142-3p and miR-150 in 4T1 and TUBO cancer cell lines and tumor tissues. Additionally, these LNA-anti-miR-142-3p-carrying exosomes homed to and were retained at tumor sites [90]. Researchers have also developed specific therapeutic peptides to anchor to exosome surfaces, allowing the promotion of exosome functions and direction of exosomes to targeted cells [91,92]. Therefore, it is possible to load a therapeutic agent into exosomes for the treatment of diseases, including OA.

Given the natural transport of bioactive molecules by exosomes, whether exosomes can be used to carry targeted drugs is a major concern, and it is important to develop exosomes for use in precision medicine approaches. Exosomes may possess targeting behavior and display a preference for a particular cell type or tissue. For example, exosomes tend to be taken up by the same cell type that secretes them, and the incubation of exosomes with a particular cell type may increase the internalization of exosomes into the cells [93,94]. Exosomes bearing specific surface receptors derived from immune cells or MSCs bind tumors with the same tropism as their parent cells [95,96]. The “cell specificity” of exosomes is also determined by the phagocytic activity of the recipient cells. For instance, Feng et al. analyzed eight cell lines and showed that exosomes are internalized by phagocytic cells (such as THP-1 monocytes or mouse Raw264.7 macrophages) via the phagocytosis mechanism [97]. Macrophages are phagocytic cells that specialize in removing dead cells, cellular debris, and foreign substances via phagocytosis. Thus, their high phagocytotic capacity may be the reason for their highly efficient internalization of EVs. These theories, however, require more investigation, both in vitro and in vivo. Current genetic and chemical engineering technologies can be used to improve the targeting characteristics of exosomes. The introduction of targeting ligands on the exosome surface, such as a viral glycoprotein peptide that binds to the membrane or anchoring proteins, can be performed by manipulating parental cells [14,91]. However, this strategy may only be appropriate for some molecules because others may be degraded or impair exosomal function, and time-consuming and modified culture conditions may be required [98]. Alternatively, postexosome insertion, in which targeting molecules are conjugated to the tail of exosomal membrane receptors [99] or directly anchored on the exosome membrane, is likely to be an effective strategy [100]. The postexosome loading approach may be attractive because researchers can control targeting molecule density and internalization [98]. Thus, we manipulate exosomes to act as smart drug carriers, allowing them to transport drugs to target sites.

Along with developing technology to manipulate therapeutic and targetable exosomes, it is necessary to optimize isolation and purification techniques to facilitate the future clinical application of exosomes. Differential centrifugation is the most common technique used to separate exosomes, in addition to various other approaches, including density gradient production, precipitation, filtration, size exclusion chromatography, and immune isolation [101]. The basic method of differential centrifugation involves several sequential steps with increasing centrifugation forces and thus presents many challenges, including low yield and purity, time intensiveness, high cost, and difficulty in standardization [102]. Additionally, this method is appropriate for low-viscosity fluids, such as cell culture media or urine, but not for small clinical samples with a small volume. Immunoisolation based on the specific binding of antibodies and ligands to separate exosomes from a heterogeneous mixture has strong specificity, high sensitivity, high purity, and yield and can be used to quantitatively and quantitatively analyze exosomes. Despite the advantages of this method (low cost, high purity and high yield), the disadvantages limit its further use for functional assays, as there is a lack of standard exosomal markers, and the association between marker proteins and the secreting cells/tissues of origin has not been fully characterized; additionally, there is an overlap of markers among EV subpopulations. This technique is also not ideal for the large-scale separation of exosomes. Other techniques based on size and precipitation present challenges in obtaining adequate purity and yield and are affected by the nature of the sample since no technique addresses all sample types, as well-reviewed by Zhu et al. and Dismuke et al. [103,104]. According to guidelines from MISEV2018, the requirement for the exosome purification level depends on the experimental question, and a combination of several techniques can be used to obtain the desired results [101]. The greatest challenge is the overlap of markers and sizes among different EV populations, making it difficult to separate exosomes from other EVs. Investigators can apply a combination of different isolation techniques or characterization methods to obtain the appropriate data based on the current consensus within the exosome research community, for example, obtaining the purity level, determining the particular sample type, or performing a functional analysis. As exploration into innovative isolation and purification technology continues, more new methods will become available to investigators for optimizing traditional technologies for specific applications.

## 11. Strategy to Localize Exosomes to Damaged OA

The significant effects of exosomes on diseases, including osteoarthritis, have been discussed. Exosomes are safe, effective, and overcome the side effects associated with other drugs that can prevent tissue damage and restore the joint. Additionally, using this type of extracellular vesicle can avoid the problem of genetic instability associated with cell-based therapies. Although both synthetic particles and exosomes can protect their cargo from degradation, natural exosomes can avoid toxicity and immunogenicity due to their natural biocompatibility and higher chemical stability [105], while lipid nanoparticles are highly immunogenic [106]. Moreover, exosomes more easily cross biological barriers, such as the blood-brain barrier [105]. Murphy et al. demonstrated that both exosomes and synthetic lipid nanoparticles successfully delivered sgRNA into HEK293T cells, but the efficient delivery was at least twice as high by exosomes as by lipid nanoparticles [107]. Additionally, the author showed that exosomes were rapidly and highly taken up by recipient cells, which may explain the more efficient transfer of sgRNA by natural exosomes. This finding is consistent with those of a previous investigation by Schindler et al. in which HEK293-derived exosomes carrying doxorubicin were taken up rapidly by HEK293 cells and delivered more doxorubicin into the cytosol than other liposome-loaded doxorubicin formulations [108]. Alternatively, exosome-mimicking liposomes were more readily taken up than synthetic liposomes by A549 cells and HUVECs [109]. This phenomenon has also been reported by Reshke et al., who found that siRNA-bearing exosomes could knock down siRNA-mediated target genes in the liver, intestine, and kidney glomeruli in vivo at a dose at least 10-fold lower than that required for synthetic lipid nanoparticles [110]. This suggests that the unique cellular lipid composition may enhance the internalization and intracellular delivery efficiency of natural vesicles. However, the natural loading of therapeutic molecules into exosomes is passive, which may limit their capacity to activate cell functions [107]; thus, more studies are required to develop active techniques to load therapeutic molecules into exosomes.

However, the remaining concern is whether exosomes persist at the injected site of the joint. Direct injection of exosomes into the synovial joint may be beneficial for the localization of exosomes, but whether the injected exosomes remain in the joint and remain functional is unknown. Several suggestions have been proposed to address this, including combining exosomes with a suitable biomaterial, such as a hydrogel or scaffold [111]. Some studies indicated that exosomes combined with hydrogel sponges could accelerate wound healing and regeneration in animal models because hydrogel sponges could improve the stability of exosomes and control the release of molecules from exosomes depending on changes in the damaged joint [112,113,114,115]. Liu et al. designed a hydrogel glue (comprising *o*-nitrobenzyl alcohol moieties, modified hyaluronic acids and gelatin) to retain exosomes released from hiPSC-derived MSCs [115]. After 14 days, the hydrogel retained over 90% of the exosomes. Importantly, the exosome-hydrogel could attach to the lateral cartilage and penetrate into the subchondral bone to form a seamless interface in the rabbit articular cartilage defect model [115]. Recently, Yang et al. developed a hydroxyapatite-embedded hyaluronic acid alginate hydrogel system to retain human UCMSC-derived exosomes [116]. The hydrogel system gradually released 71.2% of exosomes into the culture environment by day 14. Additionally, this exosome-hydrogel system could promote bone regeneration in a model of bone-defective rats by increasing new bone deposition and formation and neovascularization [116]. Exosomes can also be designed to adhere to the scaffold surface to sustain delivery and improve the targetability of exosomes into the treated joint [117]. An ideal system of biomaterials integrated with exosomes to ensure their functions for OA treatment requires effective retention and gradual release of exosomes, significant facilitation of exosome functions, and accurate filling of irregularly shaped tissue defects. While methods for localized administration remain under development, direct intra-articular injection and exosome-biomaterial combination are appropriate for introducing exosomes into the damaged joint to precisely localize EVs and maintain their long-term effects.

## 12. Discussion

Considering therapeutic EVs with a focus on OA management, we have highlighted the current therapeutic potential of EVs released from MSCs for targeting the various biological processes of OA. MSC-derived EVs are a potential alternative candidate to MSCs for OA treatment due to their advantages of biocompatibility, similar effectiveness to parental cells, and ease of overcoming the biological barrier. MSC-derived EVs play roles in inflammation, ECM repair, cartilage protection, bone homeostasis, and pain relief. However, the field of MSC-derived EVs is under development and preclinical studies have been mostly carried out on small animal models; thus, further clinical trials are required to confirm the clinical application of MSC-derived EVs in OA treatment.

Intensive investigation of MSC-derived EV compositions related to OA remains poor. Only several compositions, such as exosomal proteins (CD73, Wnt5a/Wnt5b, and proteins presented in Table 1), lncRNAs (H19 and KLF3-AS1), and miRNAs (Table 2), have been reported. These factors further affect various signaling pathways, such as AKT/ERK, Wnt, PTEN/AKT, and RAN/RANK/TRAF6. Thus, although MSC-derived EVs demonstrate substantial potential for OA treatment, further investigations into the mechanisms underlying EV function are required. The effects of OA recovery based on EVs may not occur through a single factor but instead result from the overall content of the EVs, including molecules that play different roles, such as those with catabolic to anabolic activities. It is compulsory to demonstrate the correlation between the biological activity of EVs and disease outcomes. As the mechanism of MSC-derived EVs is a complex that has yet to be fully described, the US Food and Drug Administration (FDA) has advised using a combinatorial assay matrix including biological (in vitro or in vivo) or biochemical tests. Potency tests, which measure biological activity, quantification, purification, preparation and consistency, will determine whether EV drugs are released [118].

To date, the majority of studies have investigated EVs naturally secreted by MSCs, leading to inconsistency in EV cargoes due to the different cell culture techniques, cellular origins, or pathological states. The trend is moving toward synthesizing EVs carrying a predominant therapeutic molecule that will be able to direct engineered EVs for a particular disease. Additionally, engineered EVs will be precisely targeted to the appropriate cells/tissues. This direction will be useful to generate desired EV products for the treatment of diseases, including OA.

Different models for EV-based OA research have been proposed, both in vitro and in vivo. However, MSC-derived EVs have not been tested in enough large animal OA models, especially the nonhuman primate or sheep OA model, delaying the transition to clinical studies. To date, only two clinical studies registered on the website of clinicaltrials.gov for using exosomes and conditioned media from MSCs to OA (https://clinicaltrials.gov/ct2/show/NCT05060107 and https://clinicaltrials.gov/ct2/show/NCT04314661, accessed on 18 October 2021).

The clinical application of EVs has been delayed not only by the unclear mechanism but also by the challenges associated with large-scale isolation and production, including EV separation, purification, and storage, while maintaining function. Despite the fact that various techniques have been commercialized, traditional differential centrifugation is still preferred and can be used for both small and moderate sample volumes. Thus, industrial companies are enforcing the development of innovative technology to isolate EVs effectively and consistently at the industrial scale.

## 13. Conclusions

An increasing number of researchers have developed extracellular vesicles, especially exosomes, derived from MSCs for OA management. Notably, MSC-derived exosomes can regulate ECM synthesis and degradation, chondrocyte proliferation, migration and apoptosis, inflammatory modulation, bone homeostasis, and pain relief. Due to these satisfactory results, therapeutic exosomes may be an innovative medicine and a potential substitute for stem cell therapy for OA patients. Advances in current techniques promise to produce therapeutic EVs targeting OA that can overcome the disadvantages of current OA medications to reduce OA symptoms and potentially regenerate the damaged OA joint. However, previous works were mainly conducted at the preclinical level and in several small animal models. There is a lack of large OA animal models and clinical studies to pave the way to the future clinical use of exosomes as a cell-free biological product. Therefore, innovative strategies to manipulate drug exosomes and administer targeted exosomes should be developed to bring smart therapeutic exosomes to the clinic for osteoarthritis management in the near future.

## Figures and Tables

**Figure 1 cells-10-02887-f001:**
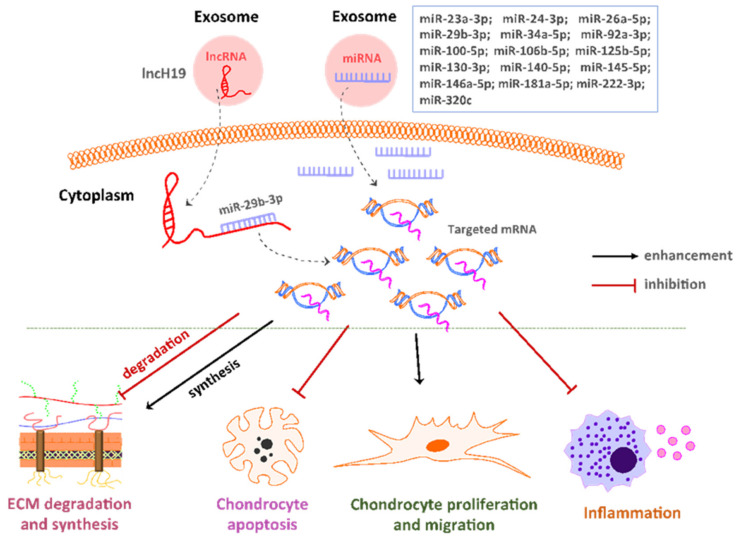
Influences of exosomal microRNAs and long noncoding RNAs on different biological activities of osteoarthritis.

**Figure 2 cells-10-02887-f002:**
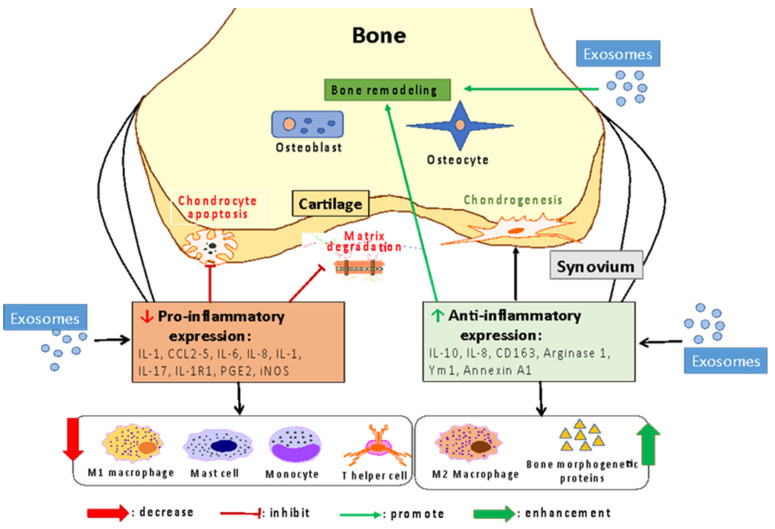
The mechanism by which exosomes promote osteoarthritis recovery through modulating inflammatory responses.

**Figure 3 cells-10-02887-f003:**
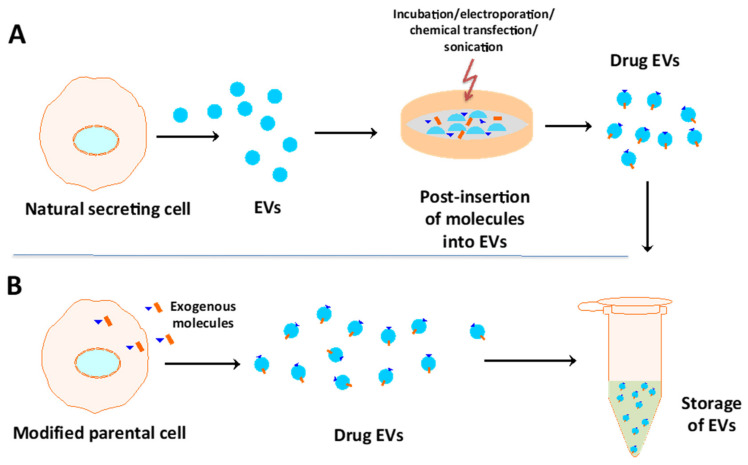
Strategies to enrich drug molecules into EVs include (**A**) the direct incubation of drugs with exosomes/EVs and (**B**) the indirect approach of inserting molecules into secreting cells that will release exosomes/EVs with more of the inserted factor.

**Table 1 cells-10-02887-t001:** MSC-derived exosomal proteins have potential in OA regulation.

Functional Proteins	Sources of EVs	Effect	Reference
TGF-β1	BMMSCs	Enhance proliferation, migration, and fibrosis of tenocytes	[29]
CD9	BMMSCs	linked to osteoclastogenesis that can promote osteoblast fusion and bone healing	[30]
CD73	hMSCs	Reduce inflammation and maintain mediate matrix homeostasis by activating AKT/ERK phosphorylation via AMP hydrolysis	[31]
Annexin A1	ADMSCs	Reduce inflammatory effects of IL-6 and restore the ECM by inducing COL II production	[32]
DKK-1	ADMSCs	Promote chondrogenesis and chondrocyte redifferentiation by blocking Wnt signaling	[33]
BDNF	BMMSCs	Increase expression of osteogenic markers and modulate bone repair process	[33]
HGF	BMMSCs	Induce osteogenic differentiation by increasing expression of osteogenic markers	[33]

**Table 2 cells-10-02887-t002:** Changes in the expression level of factors involved in ECM cartilage homeostasis under MSC-derived exosomes.

	Gene Name	Encoding Protein	Function	Reference
Increased expression level	ACAN	Aggrecan	Major ECM proteoglycan in the articular cartilage.	[20,68]
	COL2A1	Collagen type II	The main component of collagen fibril-structural backbone of the articular cartilage.	[45,48]
	SOX-9	SRY-related HMG-box-9	TF-expressed by proliferating chondrocytes that maintain cartilage ECM homeostasis.	[51,61]
	PRG4	Proteoglycan 4 (or lubricin)	Secreted by synovial fibroblasts and superficial zone chondrocytes that regulate joint homeostasis.	[69]
	COMP	Cartilage oligomeric matrix protein (or thrombospondin 5)	Structural role in endochondral ossification and the assembly and stabilization of ECM	[48]
Decreased expression level	MMP-1/-3/-13	Matrix metalloproteinases-1/3/13	Collagenase-responsible for the collagen and other protein degradation in ECM	[38,52]
	ADAMTS5	Aggrecanase-5	An aggrecanase-a proteolytic enzyme that cleaves aggrecan	[50,70]
	Runx2	Runt-related transcription factor 2	TF-promote the expression of catabolic factors to the cartilage ECM	[20,66]
	WNT5A	Wingless-type MMTV Integration Site Family, Member 5A	Activate MMPs along with reducing cartilage formation and ECM synthesis	[20]
	COL10A1	Type X collagen	Expressed explicitly by hypertrophic chondrocytes during endochondral ossification	[20]

**Table 3 cells-10-02887-t003:** MSC-derived exosomal miRNAs in OA-ECM regulation.

miRNAs	Targeted RNA	Effect	Reference
miR-23a-3p	PTEN	Upregulate P-AKT and activate PTEN/AKT signal pathway, resulting in glycosaminoglycan formation, extracellular matrix synthesis, and collagen II deposition	[51]
miR-100-5p	mTOR	Induce mTOR-regulated autophagy leading to the increase in ECM synthesis	[52]
miR-320c		Upregulate SOX9 and downregulate MMP13 expression in OA chondrocytes	[28]
miR-92a-3p	WNT5A	Suppress the activation of MMPs together with enhancing cartilage formation and ECM synthesis	[20]
miR-136-5p	ELF3	Promoting chondrocytes migration while increasing collagen II, aggrecan, and SOX9 expression and decreasing MMP-13 expression.	[71]
miR-127-3p	CDH11	Blocking the Wnt/β-catenin pathway activation, which contributes to chondrocyte damage and promotes the progression of OA	[72]

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
