# Peer review of "Mesenchymal Stem Cell-Derived Extracellular Vesicles for Osteoarthritis Treatment: Extracellular Matrix Protection, Chondrocyte and Osteocyte Physiology, Pain and Inflammation Management"

_cells, 2021, doi:10.3390/cells10112887_

Round 1
Reviewer 1 Report
In this work Nguyen and colleagues performed a literature review to evaluate the use of MSC-derived exosomes in Osteochondral (OA) applications. Overall, the text is well written and organized. The manuscript covers a broad range of topics, including 1) reasons why exosomes are potential candidates for cell-free therapies of OA (regeneration, apoptosis and inflammation), and 2) future trends of the application of exosomes (e.g. strategies to localize and/or develop therapeutic EVs).
Comparing this manuscript to other published papers on the topic (e.g. DOIs 10.1186/s13287-021-02138, 10.1016/j.mtbio.2020.100067, 10.1002/stem.2575 and 10.1002/smll.202101741), this manuscript is easier to understand and can be used as a gateway to the world of MSC-derived exosomes. Particularly, the information in sections 1-6 is organized and clear. Nevertheless, the stakes and competition from the available literature are high and some key areas in this manuscript must be improved. I believe this manuscript can be published in cells (mdpi) after the authors have addressed some of mine major concerns to their manuscript.
1- The title suggests an overview of OA, which includes cartilage, bone tissue and immune cells. However, bone tissue and immune cells are not thoroughly discussed in the manuscript. Consider changing the title and the introduction.
2- In section 1you introduced OA and clearly justified the use of exosomes in a general matter. However, I feel this section should be followed by a “physiology” section to address what stimuli lead to normal exosome production in MSCs and how is this affected in OA.
3- In section 2 I believe the reasons for using MSC-derived exosomes in should be expanded. For example, I propose reviewing the chondrogenesis process and evaluate in which steps of differentiation/maturation do MSC-derived exosomes are critical.
4- Sections 3 and 4 deal with a more mechanistical review on the effects of exosomes on ECM, and cell metabolism. I particularly like this section and propose some minor changes to it:
4.1. When discussing the effects of MSC-derived exosomes on cells, highlight the MSC source used (e.g. bone-marrow, adipose tissue, synovial).
4.2. Consider discussing the effect of MSC-derived exosomes on the production of specific ECM components, such as chondroitin sulfate and hyaluronic acid.
4.3. Consider discussing the differences between MSC-derived exosomes depending on the MSC source used (e.g. bone-marrow, adipose tissue, synovial) regarding content and surface properties.
5- In section 4 you discuss the role of exosomes on chondrocyte apoptosis and migration. How easily can exosomes permeate through cartilage vs synthetic vesicles? Are the effects then limited to surface chondrocytes or there is a general tissue stimulation?
6- In section 5 you address the effect of exosomes on inflammation. As you mention, macrophages are paramount for inflammation and different miRNAs found in exosomes and stimulate/inhibit M1/M2 types. Please further expand this section for a better understanding on the role of macrophages in OA progression.
7- Overall I found sections 7 and 8 very vague. I propose expanding some of the topics discussed here, including:
7.1. Exosomes can interact with many cell types. For example, MSC-derived exosomes can interact with endothelial cells and promote angiogenesis (DOI: 10.18632/oncotarget.16778). How is this tropism manipulated by cells and what specific strategies can be used to manipulate such? Can this affect chondrocyte health and OA progression? Consider highlighting one or 2 significant studies.
7.2. What are the main advantages of exosomes vs synthetic vesicles and why should researchers/clinicians consider the first over the second?
7.3. What strategies have been employed to integrate exosomes with biomaterials and regenerative therapy strategies for OA applications? Consider highlighting one or two examples of such studies and comment on their clinical applicability.
7.4. Consider addressing exosome glycobiology (DOIs: 10.1038/s41598-019-47760-x, 10.3390/polysaccharides2020021, 10.1080/20013078.2018.1442985) and clinical applications. There are very few examples of such topic for MSC-derived exosomes.
Author Response
We would like to thank the reviewer for your critical feedback on the manuscript. We have addressed all the comments as in the revised manuscript and attached file.

Reviewer 2 Report
Although the review plan is interesting, the two major hurdles are :
1) Need to be rephrased by a native english speaker/writer
2) Need to add quantitative data => there are nearly no quantitative data on the whole manuscript, for example we do not know at all the amount of drugs that can be encapsulated, the amount of Exosomes injected in models, the retention time in the articulation, etc). It therefore only gives vague ideas on the subject
3) the authors do not give critical and original opinions/discussions on the concept and data discussed. Apart from giving a list of papers, readers may be interested into knowing the critical opinion of authors on the literature, otherwise they can find other reviews on the subject.
Once these issues are solved, it may be reviewed to discuss other smaller issues in details

Author Response

(The authors gave the same response as above.)

Reviewer 3 Report
Specific comments
The summary of exosomal miRNAs in Table 2 seems limited and may not be updated. There should be more candidate miRNAs reported in the literature.
Apart from the discussion on EV/exosomal miRNAs and summary in Table 2, there should also be discussion of EV proteins implicated in the modulation of various cellular processes relevant for tissue repair in osteoarthritis. For instance, exosomal proteins such as CD73 and TGF-β have been reported to mediate cellular processes such as proliferation and matrix synthesis. Authors are urged to critically discuss the underlying mechanisms of MSC-EVs in cartilage repair, with reference to the specific cargo molecules including proteins and miRNAs.
There should be discussion on the current challenges related to the isolation and characterisation of EVs that requires standardization and rigor in the field. Authors are referred to the position paper (MISEV2018) of International Society of Extracellular Vesicles (ISEV) for the guidelines on nomenclature, isolation, characterisation of EVs. It is important to note that due to overlapping size range and the lack of specific markers, EV preparations including exosomes are heterogenous.
It is important to discuss the larger animal studies that are required to demonstrate the safety and functional efficacy of MSC-EVs, critical for clinical relevance and translation. On this note, there are recent larger animal studies that should be included in the review.
An important aspect of OA management is pain. Authors should also be discussing the role of MSC EV/exosomes in alleviating pain and the possible mechanisms that have been reported to date.
Author Response

(The authors gave the same response as above.)

Round 2
Reviewer 1 Report
Overall I believe the quality of the manuscript has been improved and critical analysis were made to some interesting topics. I believe the manuscript is closer to publication. Here are some suggestions (minor revisions) to further improve the manuscript:
1- My last 2nd review comment was not completely addressed. Instead, you have discussed the role of external stimuli on exosome production, which I find to be ok. Nevertheless, please add a small section discussing exosome formation and excretion by MSCs. As a reference for the layout, check the paper of Ibrahim and colleagues (DOI: 10.1146/annurev-physiol-021115-104929), section “Exosome Biogenesis and Release”.
2- The text in lines 145-149 should be reformulated to improve the meaning of the conveyed message.
3- In phrase 242-243 the term “sulfate glycosaminoglycan” should be changed to “sulfated glycosaminoglycans”. Moreover, for this very interesting study add 1) quantitative data and 2) comment on whether the cartilage produced is similar to native one.
4- I believe readers should be aware that most studies contained here are in fact in vitro. For the whole manuscript, please specify for each in vivo assay the model specie used and the origin of the exossomes (same or different specie).
5- Phrase 257 please note the phrase “…Versican, which is a chondroitin sulfate produced…”. In fact, versican is a proteoglycan rich in chondroitin sulfate. Please check this and similar misspelling in the manuscript.
6- Phrase 384-385 please consider replacing “proinflammatory/anti-inflammatory effects” by a more correct terms “pro-inflammatory/anti-inflammatory factors”.
Author Response
We would like to thank the reviewer for your positive feedback on the manuscript. We have addressed all the comments and edited the manuscript as below:
1, My last 2nd review comment was not completely addressed. Instead, you have discussed the role of external stimuli on exosome production, which I find to be ok. Nevertheless, please add a small section discussing exosome formation and excretion by MSCs. As a reference for the layout, check the paper of Ibrahim and colleagues (DOI: 10.1146/annurev-physiol-021115-104929), section “Exosome Biogenesis and Release”.
Response:
Thank you for the reviewer patient. We have further added a small paragraph on the EV formation and release as below:
“Depending on their biogenesis mechanism, EVs can be classified into three categories: apoptotic bodies, microvesicles, and exosomes. Among these, exosomes are the most attractive to researchers, followed by microvesicles, and there is not much consideration of apoptotic bodies, which are apoptosis products. The formation of exosomes starts from the inward budding of an endocytic vesicle that is then fused with early endosomes in the cytoplasm. This early endosome then develops into multivesicular bodies. Multivesicular bodies either deliver their contents to the lysosome for degradation or release exosomes into extracellular spaces by fusing their membrane with the cell membrane (34). Although the factors involved in determining the fate of multivesicular bodies have not been well investigated, several studies have shown that cholesterol-rich MVBs fuse to the plasma membrane. Additionally, the endosomal sorting complex required for transport (ESCRT)-dependent pathway or ESCRT-independent pathway has been proposed to contribute to regulating exosome biogenesis (35). In a different mechanism, microvesicles are formed by outward budding and fission of the plasma membrane. The direct budding of microvesicles occurs only at plasma membrane sites with changes in local lipid composition and phospholipid translocation. This process is completed through ARF6 and RHOA-dependent actin cytoskeleton reorganization (34).”
2, The text in lines 145-149 should be reformulated to improve the meaning of the conveyed message.
Response:
We have further edited these sentences as below:
“The tissue origin, such as bones, muscles, or brain, and healthy or pathological states of the secreting cells also influence molecules packaged into exosomes. For example, senescent cells can secrete exosomes carrying different contents, such as a low level of miR-140-3p, leading to exosomes with impaired regenerative capacity compared to those secreted by nonsenescent cells (36, 37).”
3, In phrase 242-243 the term “sulfate glycosaminoglycan” should be changed to “sulfated glycosaminoglycans”. Moreover, for this very interesting study add 1) quantitative data and 2) comment on whether the cartilage produced is similar to native one.
Response:
We have edited the sentences as below:
“Weekly intra-articular injection of 100 μL human embryonic MSC-derived exosomes (concentration of 1 μg/μL) each up to 12 weeks was also able to enhance matrix synthesis of collagen type II and sulfated glycosaminoglycans (s-GAG) (65). After 12 weeks of exosome treatment, defects showed good regeneration, with hyaline cartilage integrated completely with the surrounding cartilage and subchondral bone closely matched to the age-similar native control (65).”
4, I believe readers should be aware that most studies contained here are in fact in vitro. For the whole manuscript, please specify for each in vivo assay the model specie used and the origin of the exossomes (same or different specie).
Response:
We have gone through the whole manuscript and specified the assay for each provided information.
5, Phrase 257 please note the phrase “…Versican, which is a chondroitin sulfate produced…”. In fact, versican is a proteoglycan rich in chondroitin sulfate. Please check this and similar misspelling in the manuscript.
Response:
Thank you for this comment. We have checked and revised the sentence as below:
“ADMSC-derived EVs were abundant with versican, which is a chondroitin sulfate proteoglycan that promotes chondrogenesis and joint morphogenesis (41, 67).”
6, Phrase 384-385 please consider replacing “proinflammatory/anti-inflammatory effects” by a more correct terms “pro-inflammatory/anti-inflammatory factors”.
Response:
We have corrected this terms through whole manuscript.

Reviewer 2 Report
The paper has been vastly improved. You will find attached the pdf file with small comments to be corrected.
Niece work.

Author Response
Thank you reviewer for your positive feedback on our previous correction. We have further edited the manuscript as below:
- The title: Mesenchymal Stem Cell-Derived Extracellular Vesicles for Osteoarthritis Treatment: Extracellular Matrix Protection, Chondrocyte and Osteocyte Physiology, Pain and Inflammation Management.
- Line 22-23: we have edited the sentence as below:
“Despite this advancement, different studies employing EV treatment for OA have revealed reverse outcomes depending on the EV cargo, cell source, and pathological condition.”
- Line 103: we have edited the sentence as below:
“Recently, the potential role of glycans, which consist of a large number of monosaccharides linked through glycosidic bonds, in EVs has attracted interest from investigators.”
- Line 149-152: We have moved this information to the appropriate position and deleted these sentences.
- Line 241: we have editted the sentences as:
“Weekly intra-articular injection of 100 μL human embryonic MSC-derived exosomes (concentration of 1 μg/μL) each up to 12 weeks was also able to enhance matrix synthesis of collagen type II and sulfated glycosaminoglycans (s-GAG) (65). After 12 weeks of exosome treatment, defects showed good regeneration, with hyaline cartilage integrated completely with the surrounding cartilage and subchondral bone closely matched to the age-similar native control (65).”
- Line 499: we have edited the sentence as:
“The miR-30b-enriched exosomes were then applied to human umbilical vein endothelial cells and increased tube length by nearly 68% over that observed in the control (87).”
- Line 600: we have edited the sentence as below:
“Although both synthetic particles and exosomes can protect their cargo from degradation, natural exosomes can avoid toxicity and immunogenicity due to their natural biocompatibility and higher chemical stability (105), while lipid nanoparticles are highly immunogenic (106).”
- Line 656: We have corrected the information as below:
“However, the field of MSC-derived EVs is under development and preclinical studies have been mostly carried out on small animal models; thus, further clinical trials are required to confirm the clinical appication of MSC-derived EVs in OA treatment.”
- Line 678 – 680: We thank you very much for your information. These have been furthered added in the manuscript as below:
“However, MSC-derived EVs have not been tested in enough large animal OA models, especially the nonhuman primate or sheep OA model, delaying the transition to clinical studies. To date, only two clinical studies registered on the website of clinicaltrials.gov for using exosomes and conditioned media from MSCs to OA (https://clinicaltrials.gov/ct2/show/NCT05060107 and https://clinicaltrials.gov/ct2/show/NCT04314661).”

Reviewer 3 Report
Specific comments
Suggest to revise the manuscript title to be more concise.
In the Conclusion, authors should discuss the some of the current challenges faced by the field of MSC-derived EVs in clinical translation. These include the critical need for appropriate identity and potency parameters to ensure the quality control and release of MSC-derived EVs for clinical applications. Authors should refer to the recent position/review papers from ISEV/ISCT/SOCRATES workshops held to discuss these issues.
Critical considerations for the development of potency tests for therapeutic applications of mesenchymal stromal cell-derived small extracellular vesicles. Cytotherapy. 2021 May;23(5):373-380.
Author Response
We would like to thank the reviewer for your critical feedback on the manuscript. We have addressed all the comments and edited the manuscript as below:
1, Suggest to revise the manuscript title to be more concise.
Response:
We have revised the manuscript title as below:
“Mesenchymal Stem Cell-Derived Extracellular Vesicles for Osteoarthritis Treatment: Extracellular Matrix Protection, Chondrocyte and Osteocyte Physiology, Pain and Inflammation Management”
2, In the Conclusion, authors should discuss the some of the current challenges faced by the field of MSC-derived EVs in clinical translation. These include the critical need for appropriate identity and potency parameters to ensure the quality control and release of MSC-derived EVs for clinical applications. Authors should refer to the recent position/review papers from ISEV/ISCT/SOCRATES workshops held to discuss these issues.
Critical considerations for the development of potency tests for therapeutic applications of mesenchymal stromal cell-derived small extracellular vesicles. Cytotherapy. 2021 May;23(5):373-380.
Response:
We have further provided the information as below:
“It is compulsory to demonstrate the correlation between the biological activity of EVs and disease outcomes. As the mechanism of MSC-derived EVs is a complex that has yet to be fully described, the US Food and Drug Administration (FDA) has advised using a combinatorial assay matrix including biological (in vitro or in vivo) or biochemical tests. Potency tests, which measure biological activity, quantification, purification, preparation and consistency, will determine whether EV drugs are released (118).”
